# Supervised Contextual Embeddings for Transfer Learning in Natural Language Processing Tasks

**Mihir Kale, Aditya Siddhant, Sreyashi Nag, Radhika Parik, Matthias Grabmair & Anthony Tomasic**
Language Technologies Institute
Carnegie Mellon University
Pittsburgh, PA 15213, USA
`{mihirsak,asiddhan,sreyashn,rparik,mgrabmai,tomasic}@cs.cmu.edu`

## Abstract

Pre-trained word embeddings are the primary method for transfer learning in several Natural Language Processing (NLP) tasks. Recent works have focused on using unsupervised techniques such as language modeling to obtain these embeddings. In contrast, this work focuses on extracting representations from multiple pre-trained supervised models, which enriches word embeddings with task and domain specific knowledge. Experiments performed in cross-task, cross-domain and cross-lingual settings indicate that such supervised embeddings are helpful, especially in the low-resource setting, but the extent of gains is dependent on the nature of the task and domain.

## 1 Introduction

Named entity recognition, semantic role labelling, relation extraction etc. can be thought of as primary tasks necessary for solving high level tasks like question answering, summarization etc. However, labelling large amounts of data at this granularity is not only prohibitively expensive, but also unscalable. Given that high performance models for these tasks already exist, it is desirable to leverage them for other language understanding tasks.

Next, consider the domain adaptation setting where some domains have a lot of data, while others do not. A model for a low-resource domain would benefit from information in *expert* models trained on other data rich domains. Finally, consider the setting of cross-lingual adaptation, a common problem for personal assistants expanding to more languages. As the number of languages increases, it becomes unfeasible to obtain human annotated data. Again, the need to adapt to low-resource languages can be met by leveraging models that already exist for high-resource languages.

Motivated by the above scenarios, we propose a simple method to transfer (1) *supervised knowledge*, from (2) *multiple sources*, (3) in an *easy to implement* manner. In our approach, this knowledge is extracted from source models in the form of contextual word embeddings. We treat preexisting models as embedding extractors, which are used to extract token level representations for an input sentence. These representations are then combined via a task specific convex combination.

Unsupervised transfer learning methods such as ELMo have shown great success for a variety of tasks Peters et al. (2018). While they have the advantage of being trained on very large corpora, the training objectives are *unsupervised*. We show that in low-resource settings especially, leveraging representations from *multiple pre-trained supervised* models in related tasks, domains or languages can prove to be beneficial.

The common way of supervised transfer learning via fine-tuning can transfer information only from a single source task Mou et al. (2016). One way to incorporate information from multiple external sources is via multi-task learning Hashimoto et al. (2017); Ruder (2017). The limitations of multi-task learning are the need for labelled data for the source models, longer training times and complex design decisions (weighing the losses for each task, sampling strategies, and choice of architecture). In contrast, our plug-and-play approach is simple and does not assume availability of source model

data at training time. Finally, our approach also provides some interpretability (through the parameters of the convex combination) into which source tasks or domains are important for which other tasks and domains.

## 2 RELATED WORK

Our work aligns most with the following three directions of research.

**Unsupervised transfer learning**  Embeddings such as GloVe and FastText have become an integral part of the modern NLP pipeline Pennington et al. (2014); Bojanowski et al. (2017). Over the last year, language model based deep contextualized embedding methods such as ELMo have shown substantial improvements over their shallow counterparts, heralding a new era of word representations Peters et al. (2018).

**Supervised transfer learning**  CoVe McCann et al. (2017) and InferSent Conneau et al. (2017a) extract embeddings from encoders pre-trained for Machine Translation and Natural Language Inference respectively. Mihaylov et al. (2017) transfer low-level skills such as textual entailment, NER, paraphrase detection and question type classification into a reading comprehension model.

**Multi-source transfer learning**  In terms of modelling approach, our work is similar to Kim et al. (2017) , where the authors use multiple existing models for domain adaptation for spoken language understanding. In comparison, our work focuses not just on the domain adaptation, but also the cross-task and cross-lingual settings. In another work, Coates & Bollegala (2018) create meta-embeddings from multiple embeddings like GloVe, Fasttext etc.

## 3 APPROACH

Most deep learning models can be thought of as having an encoder $E$ and decoder $D$. For example in a Deep-SRL model He et al. (2017), stacked bidirectional LSTM constitutes $E$, while $D$ is the softmax layer. Assume $K$ existing supervised models either for different tasks or different domains $M_1, ..., M_K$ and corresponding encoders $E_1, ..., E_K$. Given a sentence of N tokens $(t_1, t_2, ..., t_N)$, we feed these tokens to the $K$ different encoders and get $K$ different representations for each token. We denote the encoder output of the $k$th model for the $n$th token by $h_n^k$. Each encoder generates representations specialized for the task, domain, or language it was trained for. Since our approach assumes no explicit information about the encoders of the model, they can be of varying dimensions and use different underlying architectures. Evidently, they would also be in different vector spaces and therefore we first use a projection layer to bring all of them in the same vector space. The parameters of these projection layers $W_1, ...W_K$ are learned along with the target model parameters. $W_k$ projects $h_n^k$ to a fixed $D$ dimensional vector $g_n^k$.

For inclusion in a downstream model, we aggregate the projection layer output of all the different source models into one vector. Several aggregation schemes can be employed : pooling, convex combination, attention etc. We choose the simple yet interpretable convex combination approach, as described below.

**Convex Combination**: This technique is similar to one used by ELMo Peters et al. (2018). We use a softmax normalized weight $s_k$ corresponding to each of the different representations of the word, add them up and use a scalar parameter $\gamma$ that scales up the whole vector. The embedding $O_n$ for the $n$th word comes out to be:

$$O_n = \gamma \sum_{k=1}^{K} s_k \, g_n^k$$

This approach adds $K + 1$ trainable parameters to the model. An advantage of combining the representations in this manner is that the size of the embedding is fixed irrespective of the number of source models used.

Once we get a combined representation, it can be used in the target model just like any other embedding. In our experiments, we concatenate these embeddings with traditional GloVe or ELMo embeddings.

## 4   EXPERIMENTAL SETUP

We use the proposed supervised contextual embeddings along with GloVe and ELMo embeddings in three knowledge transfer settings.

**Cross-task transfer**   In this setting, we transfer knowledge to a target task from models trained on multiple source tasks. We transfer into Semantic Role Labeling (SRL) task using Constituency Parsing (CP), Dependency Parsing (DP) and Named Entity Recognition (NER) as source tasks. The choice of SRL as a target task, with source embeddings from CP, DP and NER models, is inspired by the popular use of explicit syntactic parsing features for SRL. We use OntoNotes 5.0 Pradhan et al. (2012) dataset to train the SRL target tasks. We use the stacked alternating LSTM architechture for SRL as per He et al. (2017). On the source side, the DP model is based on Dozat & D. Manning (2016) and CP on Stern et al. (2017). For most of the source models, we use off-the-shelf, pre-trained models provided by AllenNLP [1]. We refer readers to Peters et al. (2018) for further description of model architechtures for the various tasks.

**Cross-domain transfer**   Here, we study the applicability of our method in the cross-domain setting. The target task is same as the source tasks, but instead, the domains of the source and target models are different. For this set of experiments, our task is NER and we use the OntoNotes 5.0 dataset which comes with annotations for multiple domains. Though NER is an easier task, we chose it as the target task for the cross-domain setting as even state of the art NER models may perform poorly for a data-scarce domain. We choose the target domain as web blogs and the source domains are newswire, broadcast conversation, telephone conversation, magazines and broadcast news. Note that the samples in the validation and test sets are also limited to the web blogs domain only. We use an LSTM-CRF architechture with 1 LSTM layer for NER as per Peters et al. (2017).

**Cross-lingual transfer**   From the CoNLL shared tasks, we obtain NER datasets for English, Spanish, German and Dutch Tjong Kim Sang & De Meulder (2003). We consider two scenarios with German and Spanish as the target languages and the remaining 3 as source languages. To facilitate the input of sentences into models from other languages with different scripts, we rely on cross-lingual embeddings provided by MUSE Conneau et al. (2017b). The NER model architechture is the same as the one used for the cross-domain experiments.

To study the effectiveness of our approach in the **low resource setting**, in addition to the full datasets, we also run experiments on smaller training subsets. Similar to Mulcaire et al. (2018), we create random subsets of 1,000 and 5,000 samples to simulate a low resource setting. In all the aforementoiend settings, the source task models are trained on their complete datasets.

**Hyperparameters**   We use the Adam optimizer (lr=0.001) for all our experiments. We run our target models for 50 epochs in SRL tasks and 75 epochs for NER tasks. Batch size is kept at 8 for the 1k data setting and 16 for 5k data setting. The dimensions of the GloVe and ELMo embeddings are 100 and 1024 respectively. The output dimension of the projection layer in all settings for supervised embeddings is 300.

## 5   RESULTS AND DISCUSSION

Cross-task SRL results (with GloVe and ELMo in 1k, 5k and full data settings) have been tabulated in Table 1. Table 2 has the results for cross-domain NER and Table 3 shows the results for cross-lingual transfer on NER. All the reported numbers are F1 scores.

**Cross-task SRL**   With GloVe embeddings, adding the supervised embeddings gives us significant improvements in F1 scores $\sim$ 5% for 1k and $\sim$ 7% for 5k examples. When we use the entire dataset, adding supervised embeddings provides no performance gains. Examining the learned source task weights in the 1k setting, we find that weights for CP, DP and NER have values 0.41, 0.41 and 0.18 respectively which shows that SRL benefits greatly from syntactic tasks like CP and DP. This is in

---

[1]https://allennlp.org/models

|  | #samples=1k | | #samples=5k | | #samples=all | |
| --- | --- | --- | --- | --- | --- | --- |
|  | Dev | Test | Dev | Test | Dev | Test |
| Glove | 32.30 | 33.02 | 44.98 | 46.19 | 77.62 | 77.87 |
| GloVe +Ours | **37.40** | **38.27** | **52.11** | **53.05** | 77.83 | 77.94 |
| ELMo | 44.69 | 45.34 | 58.30 | 58.79 | **82.68** | **82.58** |
| ELMo +Ours | **49.59** | **50.36** | **63.30** | **63.84** | 82.50 | 82.54 |

Table 1: Performance of cross-task transfer on SRL (samples=all includes 280K samples)

|  | #samples=1k | | #samples=5k | | #samples=all | |
| --- | --- | --- | --- | --- | --- | --- |
|  | Dev | Test | Dev | Test | Dev | Test |
| GloVe | 45.30 | 45.52 | 53.50 | 56.67 | 59.75 | **66.23** |
| GloVe +Ours | **50.18** | **49.64** | **55.49** | **60.56** | **61.16** | 65.51 |
| ELMo | 45.06 | 45.57 | 56.43 | 57.68 | 59.58 | 64.20 |
| ELMo +Ours | **48.18** | **48.56** | **56.53** | **57.94** | **60.36** | **65.19** |

Table 2: Performance of cross-domain transfer on NER (samples=all includes 17K samples)

|  | German | | | | | | Spanish | | | | | |
| --- | --- | --- | --- | --- | --- | --- | --- | --- | --- | --- | --- | --- |
|  | #samples=1k | | #samples=5k | | #samples=all | | #samples=1k | | #samples=5k | | #samples=all | |
|  | Dev | Test | Dev | Test | Dev | Test | Dev | Test | Dev | Test | Dev | Test |
| MUSE | 58.12 | 57.48 | 69.85 | 67.49 | 74.53 | 71.98 | 68.05 | 71.61 | 80.48 | 82.60 | 82.01 | 82.91 |
| MUSE +Ours | **64.85** | **61.60** | **73.06** | **70.62** | **75.56** | **72.96** | **69.23** | **74.59** | 80.48 | **82.76** | **82.11** | **84.22** |

Table 3: Performance of cross-lingual transfer on NER (samples=all is 12K for German and 27K for Spanish)

agreement with SRL state-of-the-art models Strubell et al. (2018) and Marcheggiani & Titov (2017) which rely on syntactic features.

When we replace GloVe with ELMo representations, we see that the baseline model improves by over ∼ 13%, showing that ELMo representations are indeed very strong. But adding supervised embeddings in the 1k setting further improves upon the ELMo baseline by over ∼ 5%. A similar improvement of ∼ 5% can be seen in the 5k setting as well. Our model shows comparable performance as the baseline when we use the entire dataset. These results suggest that the proposed supervised contextual embeddings further bring about improvements over already strong language model features in a low-resource setting. This reinforces the learning that when sufficient data is available, supervised signals do not provide information that the model cannot learn by itself from the data alone.

**Cross-domain NER** Supervised embeddings provide an impressive 4% improvement over the GloVe baseline with both 1,000 and 5,000 samples. Even when we replace GloVe with ELMo, we see an improvement of 3% , indicating that the benefits of using knowledge from other domains is orthogonal to what ELMo can offer. However, the gains vanish when the full dataset is used, suggesting that knowledge from other domains is particularly useful in the very low-resource setting. However, if sufficient data is available, the model has enough resources to build upon generic word embeddings. It is also interesting to note that for this dataset, GloVe based models outperform their ELMo counterparts. This is probably due to the mismatch in the data used to train ELMo (formal language from the 1 billion word corpus) as opposed to the NER dataset which consists of informal language used in web blogs.

**Cross-lingual NER** We observe substantial gains by exploiting information present in other languages. For both German and Spanish the performance gains are highest when number of samples is 1,000 , thus validating the suitability of the proposed method for transfer to very low-resource settings. Even when full dataset is used, we see gains over 1% for both languages.

## 6 CONCLUSION AND FUTURE WORK

We propose supervised contextual embeddings, an easy way to incorporate supervised knowledge from multiple pre-existing models. We perform experiments in the cross-task, cross-domain and cross-lingual setups and find that the proposed embeddings are particularly useful in the low-resource setting. Our work points to the potential of such embeddings in various downstream tasks in different transfer learning settings. Future work includes incorporating more tasks, domains and languages, and understanding the relationships among them. These explorations would build towards our larger vision of building a more complete taxonomy of transfer learning dependencies among NLP tasks, domains and languages.

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
