# OpenReview forum: "Supervised Contextual Embeddings for Transfer Learning in Natural Language Processing Tasks"
_ICLR.cc/2019/Workshop/LLD — LLD 2019_

### Official Review · AnonReviewer1 · 2019-04-06
**Very good paper. Clear and well motivated approach. Detailed experiments and discussion.**

**Rating:** 4
**Confidence:** 2

**Review:**

Summary
The paper proposes a technique to transfer knowledge from trained models to task/domains/languages where such models cannot be trained due to the lack of training data. This is done in contrast to using only pretrained embeddings such as GloVe or ELMO which are trained in an unsupervised manner, and independently from any downstream task.
The proposed approach assumes the existence of pre-trained models either for different tasks to the targeted one, or the same task but on different domains or languages. All these models are trained in a supervised manner where the last layer in the model is an output layer (before applying a Softmax). The authors project the feature representations that are generated from the pre-trained models using the Convex Combination method to unify and combine the features from all the models. Finally, they concatenate the resulting representation with either GloVe or ELMO.
In their experiments on different tasks, domains, and languages they showed that the proposed approach achieves an increase in the F1-Score ranging between 5%-7% compared to only using the pretrained embeddings.

Pros.
1.	The experimental setting fits the workshop. Specifically, training a model on limited data where the results are on par with models trained on all the available data.
2.	The proposed approach is sound. Mainly, where it does not require retraining models that were used for other tasks/domains/languages.
3.	The experiments are designed to cover the different knowledge transfer settings: different tasks, the same task but different domains, and the same task but different languages.
4.	Overall, the paper is clear and well-written, and the authors provided a good discussion on the results. Furthermore, the results are well-presented and easy to interpret.
5.	From a reproducibility perspective, the authors provided all the experimental details in their paper.

Cons.
1.	In the Related Work section, it is not clear how the work under Supervised Transfer Learning is different from this work.
2.	It is not clear how the GloVe representation was combined from the word level to the document/sentence level. If it was averaged for all the words, please mention that in the Experiments Setup section.

Further comments.
In addition to the proposed future work provided by the authors, I suggest the following:
1.	In addition to only using pretrained embedding as a baseline, compare the performance of your model to the performance of SOTA of each task but with limited data.
2.	Run the experiments on multiple subsets of the data (1k each) and report the average performance with SD.
3.	For the cross-task setting: would adding more (models trained on different) tasks always have a positive effect? Would having a large number of tasks eventually lead to a generic representation which is similar to simply using ELMO again?
4.	Minor formatting-related issues
•	Citation.  Please note the difference between having the author name inside or outside the brackets: XX (2018) discussed … / the model is provided in (XX 2019
•	In Sec.3 Approach, the acronym SRL was used without providing the full name (Semantic Role Labelling is mentioned in the first line of the introduction).

---

### Official Review · AnonReviewer2 · 2019-04-09
**Missing fine-tuning experiments**

**Rating:** 2
**Confidence:** 2

**Review:**

The authors propose a method for combining encoded representations from pretrained supervised models for transfer learning. The authors argue that in low-resource settings, the combined representations can transfer useful supervision signals that unsupervised representations cannot. The authors evaluate their work on an SRL task (cross-task) and NER (cross-domain and cross-lingual).

The authors present a complete piece of work, from a summary of transfer learning techniques, to a description of their convex combination technique, to their experiments. The paper follows a logical structure, and is generally well written.

The authors make a strong argument for the practicality of their technique. They give a thorough description of the off-the-shelf models and architectures they use, as well as a relatively static hyperparameter set. Assuming the relevant pretrained models are available, the projection layers and combination weights are easy to implement. The model indeed differs from ELMo in that the task-specific models are not jointly learned and thus need to be projected into the same space.

However, I'm concerned some key evaluations are missing. The most critical missing comparison is to ELMo/ULMFiT/BERT fine-tuned using the available labels for the transfer task (especially for cross-task and cross-domain). The most directly comparable one here is ELMo. The authors learn the exact same weights as ELMo fine-tuning would for their task: the decoder, the per-task weight, and the per-representation weight. It's likely that fine-tuning these alone would yield the performance gains obtained from concatenation with the proposed model. In general, a more complete ablation is needed besides [GloVe/ELMo] vs. [GloVe/ELMo] + proposed, e.g. proposed alone, GloVe + ELMo. In addition, the efficacy of the projection layers is not clear. The folk knowledge that a single linear can project unsupervised word vectors between languages likely does not apply here; a more complex projection might be needed to reconcile task-specific models. The authors give little justification for their projection approach and do not validate it experimentally.

---

### Decision · Program_Chairs · 2019-04-16
**Acceptance Decision**

Accept